# A Case for Below-Ground Dispersal? Insights into the Biology, Ecology and Conservation of Blind Cave Spiders in the Genus *Troglodiplura* (Mygalomorphae: Anamidae)

**DOI:** 10.3390/insects14050449

**Published:** 2023-05-10

**Authors:** Jessica R. Marsh, Steven J. Milner, Matthew Shaw, Andrew J. Stempel, Mark S. Harvey, Michael G. Rix

**Affiliations:** 1Harry Butler Institute, Murdoch University, Murdoch, WA 6150, Australia; 2Biological Sciences, South Australian Museum, GPO Box 234, Adelaide, SA 5001, Australia; 3Invertebrates Australia, Osborne Park, WA 6017, Australia; 4School of Biological Sciences, Faculty of Sciences, Engineering and Technology, University of Adelaide, Adelaide, SA 5005, Australia; 5Independent Researcher, Port Adelaide, SA 5015, Australia; 6Collections & Research, Western Australian Museum, 49 Kew Street, Welshpool, WA 6106, Australia; mark.harvey@museum.wa.gov.au (M.S.H.); michael.rix@qm.qld.gov.au (M.G.R.); 7School of Biological Sciences, University of Western Australia, Crawley, WA 6009, Australia; 8Biodiversity and Geosciences Program, Queensland Museum Collections & Research Centre, Hendra, QLD 4011, Australia

**Keywords:** climate change, genetics, invertebrate, Nemesioidina

## Abstract

**Simple Summary:**

Blind cave spiders of the genus *Troglodiplura* are large, enigmatic spiders, which show a number of adaptations to an underground existence, including elongated limbs and a complete lack of eyes. They are known only from the Nullarbor Plain, Australia and prior to this study the species were only known from juveniles or fragments of dead spiders. We investigated the distribution of *Troglodiplura* in South Australia, providing detailed observations of the behaviour of living adult and juvenile spiders in their natural habitat and in captivity. Given the adaptations of the spiders to an underground existence, plus the barriers to above-ground dispersal posed by the arid climate of the Nullarbor Plain, we expected species to be confined to single caves. However, our molecular evidence showed that the species *T. beirutpakbarai* was distributed across multiple caves in South Australia, each separated by between 10–27 km, and that there had been connection between the caves in recent history. These findings provide intriguing evidence for subterranean between-cave dispersal mechanisms for these spiders. A number of threats to cave fauna and the fragile cave ecosystem are posed, and we recommend further research to better define the distribution of vulnerable cave taxa and their conservation profiles.

**Abstract:**

Previously described from only fragments of exoskeleton and juvenile specimens, the cave spider genus *Troglodiplura* (Araneae: Anamidae), endemic to the Nullarbor Plain, is the only troglomorphic member of the infraorder Mygalomorphae recorded from Australia. We investigated the distribution of *Troglodiplura* in South Australia, collecting and observing the first (intact) mature specimens, widening the number of caves it has been recorded in, and documenting threats to conservation. Phylogenetic analyses support the placement of *Troglodiplura* as an independent lineage within the subfamily Anaminae (the ‘*Troglodiplura* group’) and provide unequivocal evidence that populations from apparently isolated cave systems are conspecifics of *T. beirutpakbarai* Harvey & Rix, 2020, with extremely low or negligible inter-population mitochondrial divergences. This is intriguing evidence for recent or contemporary subterranean dispersal of these large, troglomorphic spiders. Observations of adults and juvenile spiders taken in the natural cave environment, and supported by observations in captivity, revealed the use of crevices within caves as shelters, but no evidence of silk use for burrow construction, contrasting with the typical burrowing behaviours seen in other Anamidae. We identify a range of threats posed to the species and to the fragile cave ecosystem, and provide recommendations for further research to better define the distribution of vulnerable taxa within caves and identify actions needed to protect them.

## 1. Introduction

Caves, with near uniform annual temperature and humidity [1], low light levels, low energy inputs, and low diversity and abundance of organisms, represent extreme natural ecosystems. Many organisms living within caves are able to also exist in epigean habitats, and are termed eutroglophiles; however, others are only able to exist strictly within caves and these are termed troglobionts or troglobites [2]. Many troglobionts have specifically evolved in response to these extreme environs and show a range of morphological features, including loss or reduction of eyes, thinning or changes in the structure of the cuticle, loss of pigmentation, and elongation of appendages [3,4,5]. The repeated evolution of such phenotypes, collectively known as troglomorphisms, across diverse lineages provides models that offer insights into the developmental and genetic basis of evolutionary change.

In Australia, troglobiont spiders have been recorded from a number of families, including Anapidae [6]; Mysmenidae [7]; Pholcidae [8]; Ctenidae [9,10]; Desidae [9]; Gradungulidae [11]; Zoropsidae [10]; Linyphiidae [12]; Stiphidiidae [9]; Symphytognathidae [13]; Trachycosmidae [14]; and, the subject of this paper, the Anamidae [15]. While all of these examples are regarded as troglobionts, for many this is inferred from their location in cave systems, with only some actually showing evident troglomorphic features. Of these, the genus *Troglodiplura* Main, 1969 (Anamidae), endemic to the Nullarbor Plain or Nullarbor Divide, is clearly troglomorphic and is the only lineage of troglomorphic Mygalomorphae [15] known from Australia. The Nullarbor Plain (part of the biogeographic Nullarbor Divide) is an extensive arid landscape in southern Australia [16], covering an area of around 200,000 km^2^, and containing hundreds of caves [17]. The region is remarkable for its harsh environment, with a mean annual rainfall of 250 mm near the coast, temperatures spanning from a minimum of around −2 °C in winter to above 40 °C in summer [18]; inland there is a conspicuous lack of large woody vegetation and therefore shade. The cave environs provide a stark contrast to the extreme climes of the surface, with many caves recording near uniform annual temperatures and relative humidity deep in the cave systems, with temperatures stable around 16 °C to 18 °C depending on their locality, and relative humidity ranging between 50% and 90%, depending on the cave structure [1,19].

Spiders of the infraorder Mygalomorphae have a range of traits that make them especially vulnerable to threatening processes, such as habitat specificity, reduced physiological tolerance to changes in abiotic factors (for example sensitivity to desiccation), longevity, low vagility and a tendency towards short-range endemism, and as such are recognised for their conservation significance [20,21]. *Troglodiplura* are large spiders, showing a number of adaptations to a hypogean existence, including complete loss of eyes or eye spots, and elongate appendages [12]. Species of *Troglodiplura* are enigmatic both biologically and ecologically; prior to this study, all mature specimens in collections were known from only fragments of exoskeleton, with fresh specimens having only been collected for one species, *T. beirutpakbarai* Harvey & Rix, 2020, all of which were juveniles.

Troglomorphic adaptations can render troglobiont organisms unable to survive in the epigaeic environment [22] and so restrict above-ground dispersal from cave systems. Thus, in the absence of subterranean dispersal, cave systems can act as habitat islands which present allopatric barriers to gene flow [23,24]. Organisms restricted to such isolated systems are typically characterised by extreme genetic divergence between neighbouring sub-populations, and high levels of species endemism to particular cave systems [25,26,27,28,29]. Several examples of extremely limited dispersal have been found for spiders of the family Nesticidae [30,31], for example the troglomorphic *Nesticus barri* Gertsch, 1984 [27], which had no shared *COI* haplotypes between individuals from caves at distances of greater than 12 km [28]. Similar patterns of isolation were found for the troglomorphic spider *Troglohyphantes vignai* Brignoli, 1971 (Linyphiidae) of the Italian Alps [32], and in *Telema cucurbitina* (Wang & Lee, 2010) (Telemidae) in South China karst [33], suggesting highly restricted or no dispersal. Even for spiders with a broad range, such as the troglomorphic spider *Parastalita stygia* (Joseph, 1882) (Dysderidae), which occurs across an area of 240 km^2^ in the Dinarides mountain range in the Balkans, molecular analyses revealed deep population structuring and pronounced patterns of isolation, suggesting any dispersal is highly limited [34,35]. The foregoing case studies correspond to a “caves as islands” model [36,37]. In contrast to this model, some cave systems have variable amounts of physical linkage between caverns including via lava tubes, meso- or micro caverns, and subterranean pathways in cavernicolous limestone and/or groundwater, which may act as pathways through which troglobiont organisms can disperse [22,23]. Dispersal potential through these pathways is influenced by the presence of stratigraphic or fluvial barriers to dispersal, by the organism’s vagility, or relative dispersal ability [22,23,38,39], and by its level of troglomorphism, with more highly troglomorphic species having a lower dispersal ability [32,34]. Subterranean dispersal is more commonly recorded in stygofauna and aquatic troglobionts than in terrestrial troglobionts, where water dwelling organisms may face fewer barriers to dispersal [27,38,39,40,41]. Despite being less common, evidence of subterranean dispersal of terrestrial troglobionts has been recorded for a range of invertebrate taxa, including cave beetles of the tribe Trechini (Carabidae) in North America [38], and of the tribe Leptodirini (Leiodidae) in the Pyrenees [41,42,43]. In arachnids, evidence of subterranean dispersal was used to explain the distribution of the troglomorphic pseudoscorpion *Protochelifer naracoortensis* Beier, 1968 across multiple caves, albeit separated by only small distances [26]. Arnedo et al. (2007) collected troglobiont spiders of the genus *Dysdera* Latreille, 1804 (Dysderidae) from cracks and voids between lava tubes in the Canary Islands, providing direct evidence of dispersal of spiders through micro and meso caverns [44]. Shared haplotypes between isolated cave populations of troglomorphic species of *Cicurina* Menge, 1871 (Dictynidae), provide indirect evidence of gene flow between populations [5]. Furthermore, different *Cicurina* clade members normally only found in caves many kilometres apart have appeared sympatrically, associated with certain fractures [45]. A similar pattern of shared haplotypes was seen in troglobiont species of *Neoleptoneta* Brignoli, 1972 (Leptonetidae) [46]. Where subterranean pathways exist and where organisms are capable of dispersing, it is conceivable that instead of acting as isolated islands, caves can be viewed as a part of a broader, but still restricted, network of connected fissures and cracks, with ample potential dispersal pathways for fauna [47].

Cave environments are fragile ecosystems, which face a range of threats from anthropogenic sources including mining, groundwater extraction and contamination, impacts from above-ground development, and climate change [48]. Theoretical modelling predicts an increase in cave temperatures with increasing global temperatures, albeit with a lag time between surface increases and hypogean increases [49,50,51,52,53]. Having evolved in a hypogean environment, with more or less stable temperatures and humidity, it is likely that troglomorphic organisms have a substantially reduced tolerance to abiotic changes to the cave environment [5]. This, coupled with the limited dispersal potential of troglomorphic organisms [54], and the challenges associated with dispersal to other subterranean environs, means troglomorphic species are at a heightened risk of climate-change-induced extinction events [52,55]. The extinction risks facing troglomorphic mygalomorph spider lineages, such as *Troglodiplura*, are likely compounded by the ecological and life history traits of mygalomorph spiders more generally and such species can be considered of prime conservation concern [20,56].

Whilst surveys of the Nullarbor Plain to date have revealed a biologically intriguing, and vulnerable endemic biota (Table 1), many of the caves are yet to be surveyed. In this study we present data on surveys conducted in some South Australian caves of the Nullarbor Plain, with notes on the ecology, biology, and conservation of *T. beirutpakbarai*, and discuss findings in relation to the likely conservation status of other troglobiont spiders. Given that *Troglodiplura* is highly troglomorphic, combined with the inherent tendency towards low vagility and short-range endemism shown by mygalomorph spiders, and the barriers to dispersal presented by the harsh, arid surface environment, we hypothesise that above-ground dispersal of *Troglodiplura* is severely (if not entirely) restricted and dispersal in total is low, supported by the absence of any historical collections of *Troglodiplura* in non-karst environments. Given this, we predicted that populations of *Troglodiplura* from neighbouring caves would exhibit high levels of genetic divergence.

**Table 1 insects-14-00449-t001:** Characteristics of caves where *Troglodiplura* have been previously collected. ** indicates results collected during the present study.

Species	Cave	Geomorphic Characteristics	Habitat Where *Troglodiplura* Observed
*T. beirutpakbarai* Harvey & Rix, 2020	N-253 Eagles Rest Cave (SA)	Complex cave ca. 250 m long under calcrete pavement. Single roof hole leads to a spacious chamber 60 m × 40 m × 12 m high and to a maze of passages. Ephemeral stream with ponding and significant speleothem development (calcite and halite). Some sediments have not received water ingress for some considerable time (Milner, pers. obs.).	** Adult: dark zone on sediments. ** Adult desiccated fragments: dark zone on rocky boulder slope. ** Juveniles: diffused daylight zone near entrance, also on sediments in dark zone near live adult population.
*T. challeni* Harvey & Rix, 2020	N-83 Old Homestead Cave (WA)	Highly complex cave with over 20 km passages. Cave is now very dry with desiccated sediments and rock piles. Abundant scattered speleothem development including calcite, gypsum, and halite. The cave shows geological evidence of former extensive ponding of still water (Milner, pers. obs.).	Dark zone up to 2 km from entrance [15]
*T. harrisi* Harvey & Rix, 2020	N-327 Encompassing Cave (WA)	Complex cave, over 1 km long with mostly low passages. Calcite and other speleothem development. Tree roots penetrate the cave [57]	Habitat shared with *Tartarus* sp. in a sealed dark zone. Temperature 17.8–18.2 °C and humidity 80–90% [57]
*T. lowryi* Main 1969	N-58 Roaches Rest Cave (WA) (Figure 1)	Large collapse chamber under doline lip with two roof holes. Cave is a 30° inclined chamber 90 m × 50 m × 13 m high. The cave floor is generally rocky, with reflected daylight reaching most aspects of the cave (Milner, pers. obs.).	Dark zone [15].
*T. samankunani* Harvey & Rix, 2020	N-49 Pannikin Plains Cave (WA)	Overhanging doline with very steep talus slope descends to the water table. Several kilometres of underwater passage development leads to air chambers accessible only by cave divers [58]	Dark zone in air chamber ca. 3 km north of cave entrance [15].

## 2. Materials and Methods

### 2.1. Study Area

The Nullarbor Plain, located in southern Australia, is a vast, approx. 240,000 km^2^ arid limestone plateau that constitutes the world’s largest area of exposed karst, and hosts thousands of shallow cave and karst features [17,58]. The study area is located in the South Australian part of the IBRA 7.0 Nullarbor bioregion (Figure 2) [59]. Caves and karsts in Australia are numbered using the Australian Speleological Federation numbering system to aid identification [60]; in this manuscript we quote the cave number name the first time a cave is mentioned and henceforth refer to the cave by the cave number.

The majority, but not all of the shallow caves, blowholes and anastomosing tubes across the Nullarbor were formed underwater around 6 million years ago [61,62] and because of continental uplift, lowering of the water table, or regional uplift of limestone, are now full of air. The result is a horizontal plane of rubbly perforated limestone with a high volume of air space. Anastomosing tube development is observed in limestone all over the world [62,63] and on the Nullarbor it is an abundant and extensive form of perforation in the upper levels of Nullarbor limestone arising from epiphreatic solution (a zone of intense solution just below the top of the water table) [17,63]. In the subsurface of the plain there are many voids, ranging in size from small tubules through to large caves; the small tubules are irregular cavities, generally <20 cm in diameter, with narrow sinuous anastomosing tubes that follow bedding planes and joint surfaces [63] (Figure 3a–e). These honeycomb the limestone close to the surface of the plain, providing connectivity beneath the hard calcrete capstone and are connected to the surface via thousands of blowholes scattered across the Nullarbor [17].

The typical blowhole cave on the Nullarbor Plain is formed upwards to the surface from voids beneath by crystal weathering processes [64] and consists of a round vertical shaft (ranging from 0.5 m to 3 m diameter and up to 7 m depth), which may or may not intersect with chambers or passages of other cave types at depth; however, they consistently intersect with anastomosing tubes. Blowholes and associated shallow caves are distributed across the Nullarbor Plain in a 25–30-km-wide band located approx. 75 km inland and are evident in the study area [61]. The density of blowholes is up to 43 per 25 km^2^ [61]; in the study area encompassing N-253, N-6838, and N-5896, there are 138 known shallow caves and karst features [58], and an estimated 688 blowholes [61] with an average of 520 m between two karst features if randomly distributed.

### 2.2. Site Selection

The geomorphic characteristics of caves in which *Troglodiplura* spiders from the Nullarbor Plain have been collected are diverse (Table 1), with individual caves being up to 475 km apart. Many hundreds of caves on the Nullarbor Plain share features of those where *Troglodiplura* have been observed. In this study, we therefore focused surveys on caves which were proximal to, or shared similar features with, N-253 Eagles Rest Cave in which *T. beirutpakbarai* was previously collected (Figure 1) [15]. Based on the geomorphic characteristics of N-253, nearby caves with a moist and extensive sediment floor away from the daylight zone, with ephemeral ingress of water, were selected.

In order to protect these systems, following established practice with Australian cave data, we do not intend to publicly release, or generalise, cave data. Thus, these location data are considered as Category 1 under recommendations published by the GBIF Secretariat [65].

### 2.3. Survey Methods

Surveys were conducted by hand, with searches focusing on the sediment floor, cave walls, inter-rock crevices, and rock surfaces. Initial surveys were conducted using a Petzl Duo 6000k at 400 Lumens (approx.), or a Scurion 3000K at 400 Lumens (approx.) caving light attached to the caving helmet; however, to reduce the possibility of disturbance of *Troglodiplura* caused by vibration and light, later surveys employed red LED lights and swift but gentle movements to minimise vibrations. On collection, fresh specimens were transferred to 100% ethanol; fragments of spiders were stored dry. Given the conservation significance of *Troglodiplura*, collection of live specimens was conducted sensitively, and minimal specimens were collected to allow analysis of genetic diversity within a cave. Two adults (females) were collected alive from each of N-6838 (SAMA NN30812–13) and N-253 (NN 30806, NN30808). Other adults were observed or caught and released. On a subsequent entry to N-253, two already dead “mummified” females were found in N-253 and collected too (NN31721–22). Immatures were also collected from each cave. N-253 was surveyed on three dates, N-6838 on two, and the remainder of caves on one date. Survey times in each cave were variable and dependent on the size and complexity of each cave. As a rule, surveys were conducted until all suitable habitats and chambers had been inspected.

### 2.4. Site Characterisation

Relative humidity and temperature were recorded at each cave site using a Kestrel 3000 (Nielsen-Kellerman City: Boothwyn. State: PA, USA) handheld temperature, relative humidity, and wind speed instrument. The instrument was allowed to equilibrate with the cave environment for at least 30 minutes before measurements were taken during falling atmospheric barometric pressure to be therefore representative of typical environmental conditions. Descriptions were made of the general structure of each cave site, noting evidence of water and relevant characteristics.

### 2.5. Molecular and Phylogenetic Methods

The methods used to generate molecular nucleotide sequence data follow [15], with the following genes targeted as per previous phylogenetic analyses on Anamidae: cytochrome *c* oxidase subunit I (*COI*), 12S rRNA (12S), 16S rRNA (16S), Histone H3 (*H3*), elongation factor 1-gamma (*EF-1γ*), 18S rRNA (18S) and 28S rRNA (28S). We added five new specimens to the original 84-taxon dataset of Harvey et al. [15] (Table 2), including one specimen of *Aname salina* Wilson, Rix & Harvey, 2023, one specimen of *Kwongan wonganensis* (Main, 1977), and three newly collected specimens of *Troglodiplura* from three separate karst systems (N-253, N-5896, N-6838).

Bayesian phylogenetic analysis of the 89-taxon (seven gene) ‘expanded’ dataset replicated Harvey et al. (2020) [15], with 11 partitions applied to the 5373 bp matrix, and four Markov Chain Monte Carlo (MCMC) chains run for 40 million generations, with the first 10% of sampled trees discarded as ‘burn-in’ (see Appendix A). For a full breakdown of analysis software and parameters see [15,66].

## 3. Results

### 3.1. Survey Results

We conducted surveys of 23 caves and found *Troglodiplura* in three caves: N-253, from which *T. beirutpakbarai* was known prior to this study [15]; N-6838 Wedge Cave, located around 18 km NE. from N-253; and N-5896 Mitey Cave located around 10 km SW. of N-253 and around 27 km SW. of N-6838 (Figure 4, Table 3).

Caves N-5894, N-5895, and N-238 were within 20 km of N-253 and also received water from rainfall events. However, they differed from N-253 by not retaining a damp sediment floor as water sank deeper into unexplorable parts of the cave, or if there was some dampness or a flat floor, the cave was subject to external temperature and humidity close to the entrance, or the chamber was in the daylight zone. *Troglodiplura* was not observed in these caves during surveys.

**Table 3 insects-14-00449-t003:** Survey results, geomorphic characteristics, and notes on the cave environment for the caves in which *Troglodiplura* were observed. Specimens were living unless otherwise stated. Abbreviations: rh = relative humidity.

Cave	*Troglodiplura* *beirutpakbarai* Records	Geomorphic Characteristics	Cave Environment Where *Troglodiplura* Observed
N-253 Eagles Rest Cave (SA)	adult female (*n* = 2); juvenile (*n* = 11); desiccated, whole mature females (*n* = 3); fragments of exoskeleton, mature male (*n* = 4). Figure 5a,b.	Refer notes in Table 1. N-253 clearly takes a stream after rainfall events leaving ephemeral pools and calcite deposits; evidence of significant water flow and historical ponding in western passage. The deep cracks in the clay of the southern chamber floor suggest some considerable time since the last flood in that chamber.	Southern Chamber: 17.8 °C, 90% rh (8 March 2022) Western Passage where adults observed: 18.2 °C, 80% rh (8 March 2022)
N-6838 Wedge Cave (SA)	adult female (*n* = 2); juvenile (*n* = 8).	N-6838 is 18 km NE. from N-253. The collapse doline entrance rockpile descends to flat, wide, damp, hard, mud floor, ca. 10 m beneath the surface of limestone. Chamber is ca. 50 m × 30 m with a passage height of around 1.2 m (average). An occasional stream washes in with evidence of historical floods to 0.5 m deep.	17.3 °C, 95% rh (7 March 2022) 17.5 °C, 90% rh (11 March 2022)
N-5896 Mitey Cave (SA)	juvenile (*n* = 2).	N-5896 is 10 km SW. from N-253. The 6 m deep entrance tube leads to a 60 m × 40 m chamber, 2–3 m high, with a flat floor. There is evidence of significant water ingress since 2015.	18.5 °C, 76% rh (9 March 2022)

### 3.2. Phylogenetic Analysis

Phylogenetic analysis of the 89-taxon ‘expanded’ dataset (Table 2) revealed that *Troglodiplura* specimens collected from caves N-253, N-6838 and N-5898 represent separate but unequivocally conspecific populations of *T. beirutpakbarai* (Figure 6). In *COI* there were only slight differences among the caves, with a maximum uncorrected pairwise divergence of 1.5% between caves N-253/N-5896 and N-6838. Specimens from caves N-5896 and N-253 were more similar genetically, with an uncorrected pairwise divergence of only 0.3%, reflecting the closer geographical proximity of these two caves (Figure 4). However, the 16S haplotypes were identical for all three caves.

### 3.3. Intraspecific Variation

Available adult females (SAMA NN30806, NN30808, NN30812–13, NN31721–22, *n* = 6) were from two widely separated caves (N-253 and N-6838) and showed minimal variation in form and size.

Measurements were taken in mm, and given as mean (range): Carapace width 10.6 (9.8–11.7), carapace length 12.5 (11.8–13.4); Leg I: femur 12.8 (12.5–14.0), patella 6.7 (5.5–7.7), tibia 12.2 (11.8–12.9), metatarsus 12.3 (11.4–13.5), tarsus 6.3 (5.7–**6.9**); Leg IV: femur 13.2 (11.8–14.4), patella 5.2 (5.4–**5.6**), tibia 13.2 (**12.5**–14.2), metatarsus 18.5 (17.5–19.9), tarsus 5.7 (5.2–6.9). All measurements taken for N-6838 females lay within those found for N-253 females, except for three leg measurements indicated in bold type. Maxillary cuspule counts (left/right) taken from freshly collected females (*n* = 4) were higher in Cave N-253: NN30806 (38/40) NN30808 (35/35), than from Cave N-6838: NN30812 (27/25), NN30813 (33/31/).

### 3.4. Biology

Live adult spiders were observed within a few minutes of entering a chamber, and were located in the ‘open’ on the cave floor, or on the underside of overhanging rocks. Spiders were typically slow moving and easily caught, but moved quickly if disturbed. Smaller juvenile spiders were observed at the interface between the flat cave floor and the rock wall, and on the cave floor, close to entrances in the twilight zone. Juveniles were observed retreating into crevices and holes in the rock, or into mud holes. In N-6838, a spider was seen squeezing into a mud hole, with an opening of 4 mm × 6 mm, and another into a natural rock hole ca. 10 mm in diameter (Figure 7a,b). Fragments of mature male exoskeletons were located in small piles outside of crevices in the rock wall, or on top of rocks. No such piles of exoskeleton were found for females; however, two large, desiccated whole (‘mummified’) mature females and one presumably sub-adult juvenile, based on size, were observed on rock and substrate surfaces within cave N-253.

Potential prey items detected during cave surveys were cave crickets, beetles and other spiders, the three most commonly observed invertebrates detected in our surveys. One juvenile *T. beirutpakbarai* was caught whilst consuming a ground beetle (family Carabidae).

### 3.5. Conservation

Fox scats and dens were detected in two caves identified as potentially suitable habitat for *T. beirutpakbarai*; in one cave these were extensive, extending throughout accessible passages and into the dark zone. Recent damage caused by humans was also detected in one cave, with stalagmites broken and skulls of skeletal animal remains removed.

### 3.6. Behaviour in Captivity

A female *T. beirutpakbarai* from Cave N-253 and an immature from Cave N-6838 were kept in captivity in separate Perspex containers, with a layer of silt derived from N-253. Folded cardboard and plastic and glass tubes of various diameters were provided as shelters. However, the spiders almost never entered these and did not reside in them. Silk production was only observed when moulting—the immature spider from Cave N-6838 constructed a fine silk moulting web on the substrate, situated at one end of the enclosure, in which moulting occurred. Moulting occurred ‘out in the open’, with the spider lying on its back. Posture when standing still varied. Sometimes the adult female would stand for long periods with her body elevated above the substrate and at other times would have her body touching the substrate. Posture did not obviously correlate with exposure to stimulation, light, time of feeding or hydration state. At times, both spiders had active periods, roaming about their enclosures, extending legs and trying to climb the smooth sides of their containers.

One adult cave cricket, *Pallidotettix* sp., ca 16 mm in body length, was offered to the adult female *T. beirutpakbarai*, which was subdued quickly and consumed. No trace of the cricket was detectable after consumption. The same spider would not accept a wild-caught *Brises* sp. cave beetle (a large tenebrionid) three days later despite repeated re-introduction of this item. It would also not accept adult or immature cultured crickets, *Acheta domesticus,* despite the offered immature crickets being only ca. 25% of the spider’s body length and provided after a month of no food. The spider appeared aware of crickets moving in its enclosure and would withdraw from where a cricket was moving. On one occasion, an offered adult *A. domesticus* cricket moved directly towards *Troglodiplura*; the spider evidently detected this at a distance > 5 cm and raised its body high above the substrate, and allowed the cricket to walk completely under it three times. In contrast to the feeding behaviour of the adult, two immature spiders accepted small *A. domesticus* crickets whenever they were offered.

Overall, the spiders made minimal movements. When evasive movements were undertaken, they were slow and conservative. Conversely, on a couple of occasions the female was accidentally exposed to minor but sudden vibrations, and it would then lurch forward rapidly and run very fast. Gentle blowing on the spider never elicited a flight response.

### 3.7. Symbiotic Mite

A new and undescribed symbiotic mite species (*Imparipes* sp.: Scutacaridae) was found living on adult *T. beirutpakbarai* specimens from both caves in which adult spiders were collected. All the mites recovered from spider bodies were found in between coxae III and IV or between both coxa IV.

## 4. Discussion

Our findings of extremely low genetic divergence (≤1.5% for *COI* and identical haplotypes for 16S rRNA) between populations of *T. beirutpakbarai* from separate caves situated between 10 and 27 km apart implies recent or contemporary gene transfer between these populations. The morphology of *T. beirutpakbarai*—the absence of eyes, their gracile body form, and large size—in combination with observed behavioural responses suggest that the species is a troglobiont, truly restricted to the cave environment. Some spiders are known to disperse aerially, through a mechanism known as ballooning; however, ballooning has never been recorded in the Anamidae and is apparently rare in Mygalomorphae, having been recorded from only few families [67]. This, in combination with the extreme troglomorphism shown by *Troglodiplura*, the tendency for low vagility of mygalomorph spiders generally, the absence of any epigean records of *Troglodiplura* anywhere in Australia, and the stark contrast between the surface climate and the cave environs of the Nullarbor Plain, indicates that subterranean dispersal provides the best explanation for the observed genetic pattern. Previous studies have found direct evidence for dispersal by spiders through subterranean meso- or micro caverns in the Canary Islands [44] and the presence of shared haplotypes between spider species in isolated cave populations [46,68], which could provide indirect evidence of inter-population connectivity and gene flow. Nonetheless, our findings of extremely low genetic divergence and thus apparently interconnected populations over distances of up to 27 km are unexpected. The structure of the cave systems of the Nullarbor Plain, and the potential for inter-cave connectivity via anastomosing tubes and blowholes (Figure 2), may be implicated in the patterns of divergence in *T. beirutpakbarai*. Each karst feature provides a possible entry point for ingress of water, and energy inputs (food); this, in conjunction with extant or historic anastomosing tube connectivity between larger caves, presents a possible mechanism for genetic exchange for *T. beirutpakbarai* in this area. Two caves where *Troglodiplura* have been observed in apparently ‘sealed’ environments, distant from surface entry points including obvious blowholes, are *T. harrisi* in N-327 [57], and *T. samankunani* in N-49 [15]. The presence of these species in apparently ‘sealed’ environments provides further evidence for a cryptic but extant pathway for the flow of energy (food) into these systems, and thus potentially for genetic exchange.

Historically, the ocean retreated for the last time around 14 million years ago, after which there was a gentle uplift of the Nullarbor region [17,69]. The climate became increasingly dry with vegetation similar to that found on the Nullarbor Plain today, except that the dry eucalypt woodland extended farther inland and probably covered much of the Nullarbor Plain [17]. A warm, wet episode occurred around ~5–3 million years ago, and the climate reached its present level of dryness around one million years ago [17]. The three caves in which we detected populations of *T. beirutpakbarai* are located in Nullarbor limestone laid down 16–14 million years ago; it was a warm sea limestone, evidenced by the large marine fossils that are embedded in the walls of each of the caves [70]. Cave development of N-253 is different from the other two caves where *Troglodiplura* were observed in that the cavernous void beneath the caprock has been created by the collapse of deeper voids in Oligocene limestone during the late Miocene, in a process observed elsewhere on the Nullarbor [35]. N-6838 and N-5896 appear different from N-253 in that they are shallow caves with flank margin characteristics, forming around 6 million years ago [61].

The ancestor of *Troglodiplura* probably entered Nullarbor cave systems sometime during or after the Miocene [15]. This entry can be accounted for by two distinct models, representing two sides of a debate in cave biology, which have implications for resolving the paradox of *Troglodiplura* diversity. Firstly, *Troglodiplura* might have evolved parapatrically based on restricted gene flow between adjacent cave-dwelling and surface-dwelling subpopulations. If so, this might have occurred at any time after caves became available to colonise (i.e., up to approx. 14 million years ago) which includes a relatively early time including a climatically subtropical period. This would correspond to the parapatric/ecological speciation model. Secondly, following the climate relict model [41], *Troglodiplura* might have evolved allopatrically as a climate relict [71]. If so, this would be likely to occur after aridification stranded an anamid ancestor in a cave, surrounded by a drying landscape, with an above-ground habitat matrix considered increasingly impermeable to dispersal. If so, troglomorphic *Troglodiplura* could have evolved later, with the pulse of aridification commencing ca. 3 million years ago a key possibility. Testing the generality of this latter model is feasible due to some highly distinct arthropods that cohabit N-253. For example, at least three mesic-adapted mite taxa in different families occur here, yet these taxa are otherwise essentially known from wet forests. They are clearly mesic-restricted and are geographically separated from known relatives by distances ranging up to 2,500 km (*Geogamasus* cf *howardi* (Ologamasidae); Castriidinychidae; *Hybalicus* (Lordalychidae)) [72,73,74]. While overlapping predictions can make these models difficult to disentangle, there are opportunities for future molecular work to infer the mode and tempo of *Troglodiplura* diversification. Outcomes of such analyses could have implications for conservation as, for under a climate relict model, intolerance of a deteriorating climate is then directly associated with *Trogodiplura* biogeography.

There are other distinct arthropods that co-occur with *Troglodiplura* which provide extra research potential for these cave systems. One unusual example that could add to our understanding of *Troglodiplura* spp is the scutacarid mite *Imparipes* sp. found upon *T. beirutpakbarai* females. Scutacarids frequently show high host-specificity and are clear candidates for co-phylogenetic analysis. They are regarded as harmless commensals that feed on fungi, with males living off-host in nest-like situations. Interestingly, the only other scutacarids known from spiders are three species also from nemesioid hosts and these also inhabit the intercoxal space between coxae III and IV. These latter scutacarid species are specific to three congeneric pycnothelid hosts from Argentina [75].

As hypothesised by Harvey et al. [15] based on morphology, *Troglodiplura* was recovered as an independent lineage within the subfamily Anaminae, here referred to as the ‘*Troglodiplura* group’ (Figure 6). Our observations taken in the natural cave environment and in captivity indicate that *T. beirutpakbarai* does not construct burrows, or utilise silk for shelters, even though these behaviours are typical of other Anamidae. *Troglodiplura beirutpakbarai* were most often first located in ‘the open’, on the surface of rocks or substrate and when disturbed retreated to small crevices in the cave wall, close to ground level. Our results indicate that the species is an opportunistic generalist (and likely cursorial) predator, feeding on mid-sized cave beetles (Carabidae), cave crickets (Rhaphidophoridae) and with juveniles accepting cultured small epigean *A. domesticus* crickets (Gryllidae) in captivity. The presence of several large, whole, desiccated mature females and a large juvenile on rock and substrate surfaces illustrates the preservative nature of the cave environment, and indicates a scarcity of scavengers (at least in areas used by large *Troglodiplura*).

Whilst our findings that *T. beirutpakbarai* is not isolated to a single cave has positive conservation implications in that the species is not an extreme short-range endemic, the known range of the species still remains highly restricted and substantial threats exist across all of its known populations. Troglomorphic organisms, which have a substantially reduced tolerance to abiotic changes to the cave environment [5], are likely to be of elevated vulnerability to fluctuations in temperatures and humidity caused by anthropogenic climate change [49,50,51,52,53], and the latter will likely present a key threat to *Troglodiplura*. In addition to these indirect climatic threats, humans potentially pose direct threats to troglomorphic spiders, both from damage to the cave environment e.g., [76], as evidenced during our surveys, and through the potential collection of specimens for the pet trade [77]. Collection of wild animals for the pet trade is a major threat to wildlife around the world and whilst data on the impacts on invertebrates are lacking [78], it likely represents a key threat [79,80], especially for large and charismatic organisms, such as mygalomorph spiders [78,81]. Access into the caves is sometimes difficult, so this may provide a barrier to some collectors, but the importance of keeping the location details of caves that contain *Troglodiplura* out of the public realm is high. Caves are fragile environments and careless visitation, or even looting, can easily cause irreparable harm to the cultural values, biological integrity, and the archaeological and palaeontological record [82].

During cave surveys, we detected a high level of evident fox activity, both in the form of scats and dens, which in one cave extended throughout accessible passages. Gut content analysis studies have shown that both foxes and feral cats consume invertebrates [83,84,85], and predation by foxes has been identified as a possible threat to mygalomorph spiders [20], indicating that foxes are likely a key threatening process for *Troglodiplura*, especially for cave systems that are easily accessible from the surface, such as Cave N-58.

This study provides important baseline data for these spiders. However, a large number of caves of the Nullarbor Plain are yet to have had biological surveys and the potential for them to harbor similarly rare, endemic, and vulnerable biota is high. Dedicated surveys for *Troglodiplura* are recommended to better elucidate the distribution of species, to detect new species, identify which are likely to be at threat and the conservation actions needed to protect them. Collection and analysis of eDNA from cave sediment has been shown to be successful in detecting arthropods in cave environs [86] and provides increased opportunity to detect these species. Additionally, the use of passive surveillance, such as camera traps, has been documented as a successful method for monitoring populations of mygalomorph spiders and for collecting ecological and behavioural data [21]. The use of non-destructive survey techniques such as these is especially important in fragile habitats, such as caves and in combination, they may facilitate the detection and subsequent long-term monitoring of *Troglodiplura* populations, providing important conservation-relevant data to aid in their protection.

## 5. Conclusions

The findings we have presented in this study provide insights into the biology and ecology of this enigmatic species of spider and open up new avenues for research and investigation. Our molecular findings of little to no divergence between individuals from apparently separate cave systems were unexpected and provide intriguing evidence for a possible below-ground dispersal mechanism for these spiders. However, this hypothesis requires testing in further studies. Caves of the Nullarbor Plain vary in their geomorphic characteristics, abiotic conditions, and biota. Understanding which combinations of cave environments and geological history correlate with the presence of rare troglobionts such as *Troglodiplura* is recommended as an important goal for both biodiscovery and conservation.

## Figures and Tables

**Figure 1 insects-14-00449-f001:**
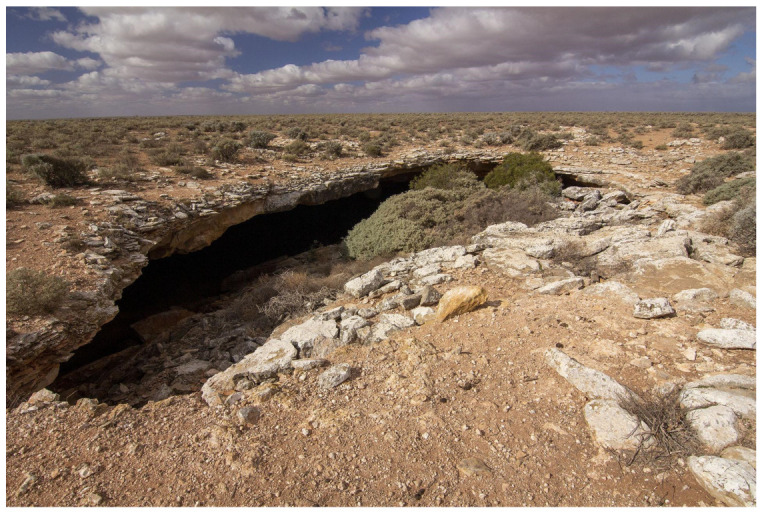
Entrance to N-58, Roaches Rest Cave, Western Australia from which *Troglodiplura lowryi* has been recorded; the entrance is around 25 m wide and up to 3 m high (Milner, pers. obs.). The species was in a dark zone of a chamber associated with a large collapse doline (Main, 1969). Photo by Steve Milner.

**Figure 2 insects-14-00449-f002:**
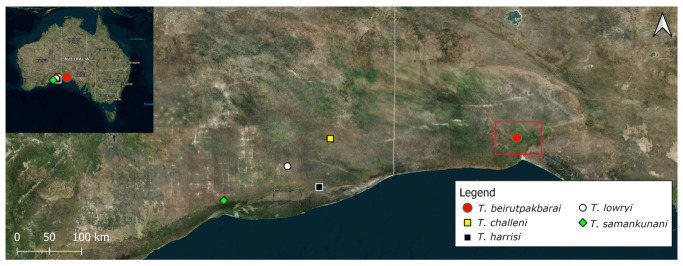
Distribution of described *Troglodiplura* species on the Nullarbor Plain. The study area is located in South Australia, outlined using a red rectangle, an area remote from other recorded *Troglodiplura* species.

**Figure 3 insects-14-00449-f003:**
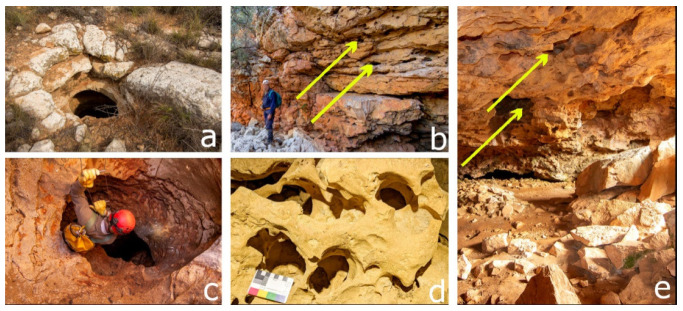
Blowholes and anastomosing tubes, photos S. Milner. (**a**) Typical small Nullarbor blowhole, approx. 40 cm diameter, 2 m deep; (**b**) exposure of anastomosing tubes in small cliff adjacent to a Western Australian Nullarbor cave entrance (arrowed); (**c**) large Nullarbor blowhole, 6 m deep; (**d**) caves may be formed by the collapse of anastomosing tubes leaving remnant blocks on the floor; (**e**) shallow South Australian Nullarbor cave showing flat floor, anastomosing tubes in wall, and remnant tubes in ceiling (arrowed).

**Figure 4 insects-14-00449-f004:**
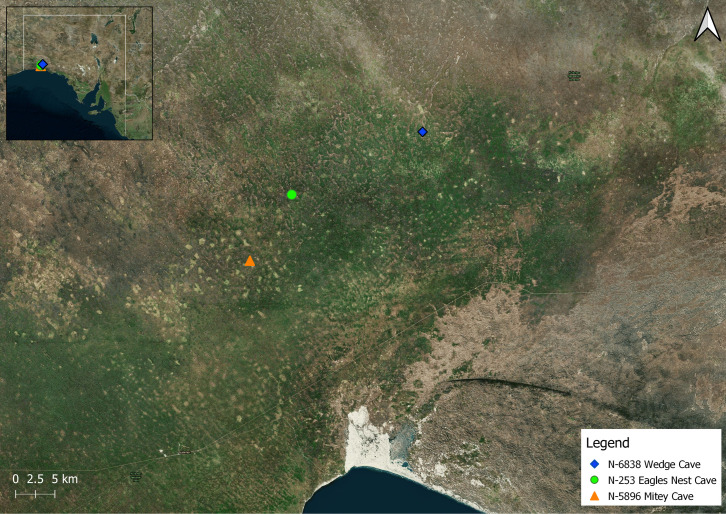
The relative positions of the three caves in which specimens of *Troglodiplura* were detected during surveys for this project—N-6838 Wedge Cave, N-253 Eagles Nest Cave and N-5896 Mitey Cave. Inset map indicates the position of the caves in South Australia. All caves were located within the study area, which was indicated by a red rectangle outline in Figure 2.

**Figure 5 insects-14-00449-f005:**
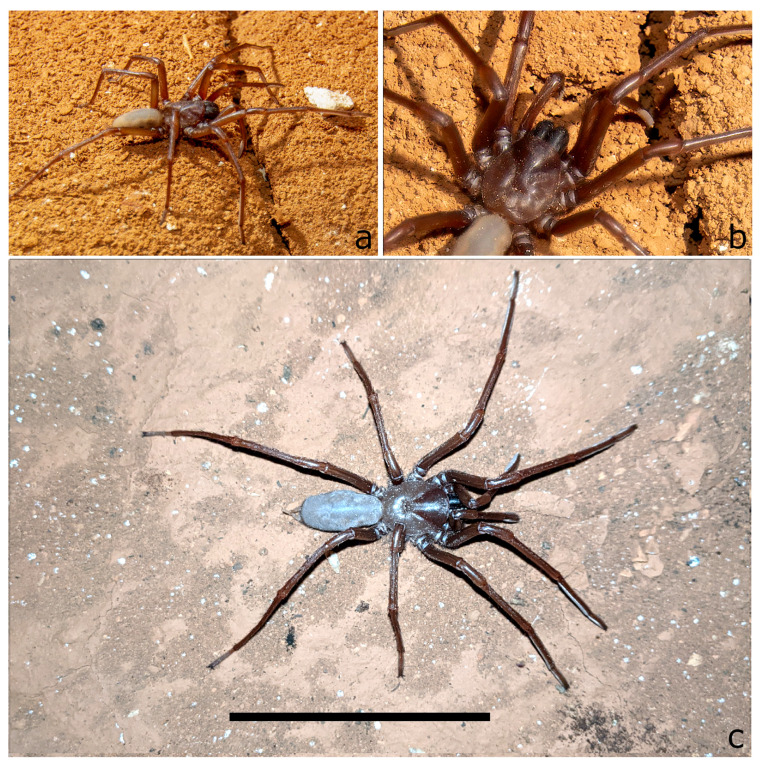
Live mature females of *T. beirutpakbarai* in the cave environment (**a**) habitus; N-253 Eagles Rest Cave; (**b**) detail of cephalothorax showing absence of eyes; N-253 Eagles Rest Cave (**c**) habitus; N-6838 Wedge Cave, scale bar = 5 cm. Photos by S. Milner.

**Figure 6 insects-14-00449-f006:**
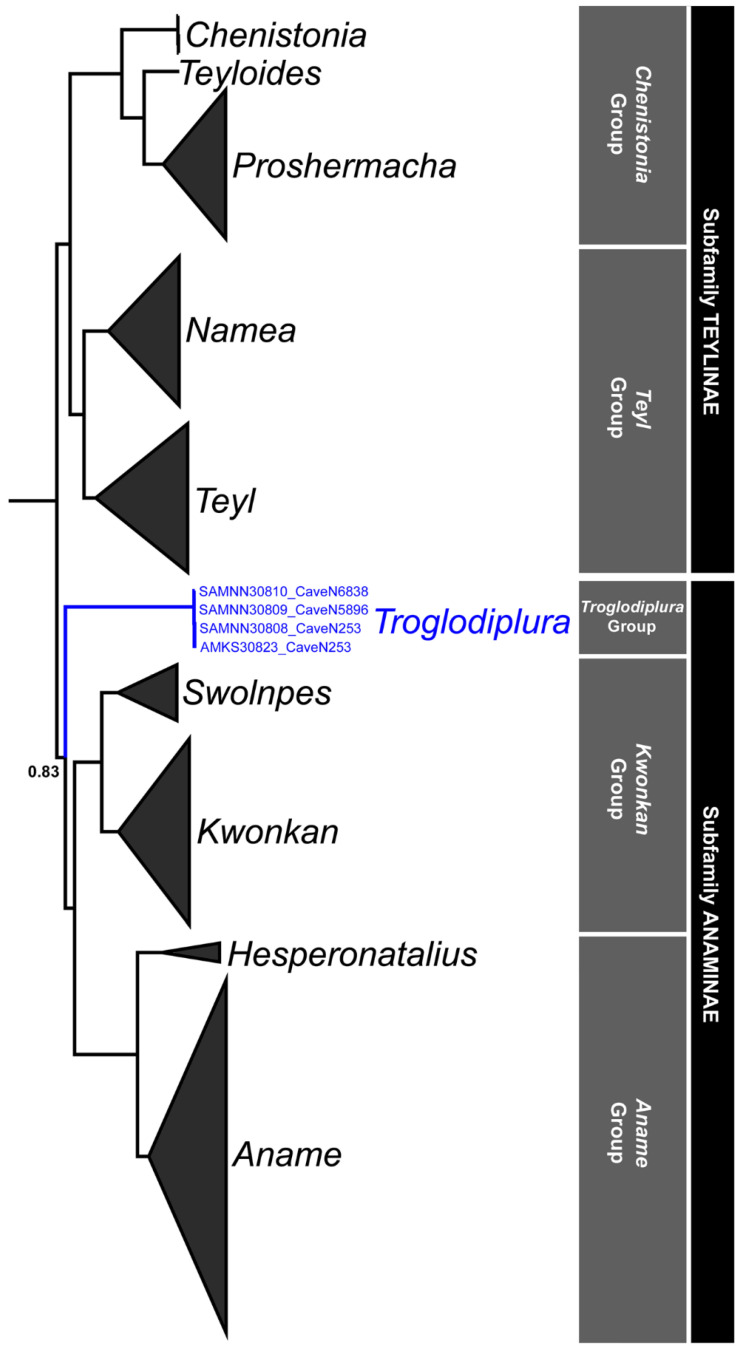
Phylogeny of Anamidae (outgroups excluded) inferred from a partitioned Bayesian analysis of the seven-gene ‘expanded’ dataset (89 taxa, 5373 bp, 50% majority-rule consensus tree), with genera and major clades of Anamidae labelled. For ‘backbone’ nodes at the level of genera or higher, posterior probabilities are all ≥0.99 unless otherwise stated. Note the position of the three newly sequenced specimens of *Troglodiplura beirutpakbarai* (SAM), and the previously sequenced specimen (AMKS30823) from Harvey et al. (2020) [15], all highlighted in blue.

**Figure 7 insects-14-00449-f007:**
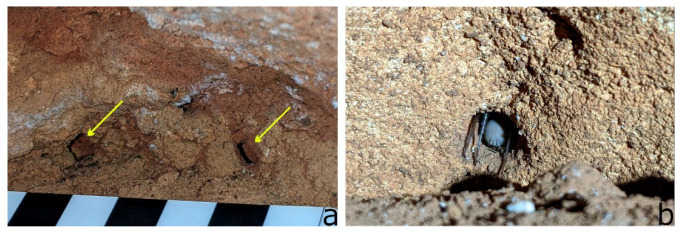
Usage of crevices in the cave environ; photos by S. Milner: (**a**) mud holes into which juvenile *T. beirutpakbarai* were observed retreating, ca. 100 mm above the sediment floor; yellow arrows indicate holes, scale bars indicate 10 mm; (**b**) spider entering natural hole in rock ca. 10 mm diameter.

**Table 2 insects-14-00449-t002:** **GenBank accession numbers for 89 specimens used in the molecular analyses.** Third-party sequences are indicated (#). AM = Australian Museum; QMB = Queensland Museum; SAM = South Australian Museum; WAM = Western Australian Museum. The square brackets indicate number of specimens.

Specimen & Depository	*COI*	12S	16S	*H3*	18S	28S	*EF-1γ*
**ARANEOMORPHAE** **HYPOCHILIDAE**							
***Hypochilus* Marx, 1888 [1] Outgroup**							
*H. pococki* (#chimera)	AF303512	KY015437	KY015913	–	KY016493	KY017132	–
**MYGALOMORPHAE—ATYPOIDEA** **ATYPIDAE** ***Atypus* Latreille, 1804 [2]**							
*A. affinis* (#ARANS000021)	KY017595	KY015315	KY015750	–	KY016328	KY016939	–
*A. karschi* (#ARAMY002283)	–	–	KY015751	–	DQ639769	KY016940	DQ680323
***Sphodros* Walckenaer, 1835 [1]**							
*S. atlanticus* (#ARAMY000643)	KY017596	KY015316	KY015752	KY018126	–	KY016941	–
**MYGALOMORPHAE—AVICULARIOIDEA (NON-NEMESIOIDINA)** **EUAGRIDAE**							
***Australothele* Raven, 1984 [1]**							
*A. jamiesoni* (#ARAMY002084)	–	–	–	–	JX069739	KY017061	JQ358731
***Cethegus* Thorell, 1881 [1]**							
*C. fugax* (WAM T129260)	KY295227	KY320448	KY320451	KY295101	KY294840	KY294963	–
**PORRHOTHELIDAE** ***Porrhothele* Simon, 1892 [1]**							
*P.* sp. indet. (QMB S111386)	–	MT280964	MT281032	MT281253	MT281107	MT281179	–
**MYGALOMORPHAE—AVICULARIOIDEA (NEMESIOIDINA)** **ANAMIDAE** ***Aname* L. Koch, 1873 [20]**							
*A. aragog* (WAM T95409)	KJ745403	KY214181	KY241234	KY241287	KY241250	KY241265	MG800219
*A. ellenae* (WAM T98890)	KJ745484	KY214186	KY241238	KY241291	KY241255	KY241270	–
*A. exulans* (WAM T121042)	MG800165	MG799896	MG799962	MG800298	MG800035	MG800112	MG800236
*A. grothi* (WAM T133820)	MN635075	MN634958	MN634775	MN635127	MN634925	MN634748	–
*A. lorica* (WAM T113826)	KJ744825	MN634945	MN634841	MN635093	MN634935	MN634733	MN635140
*A. mainae* (WAM T144398)	MN635077	MN634938	MN634859	MN635092	–	MN634735	–
*A. marae* (WAM T98424)	KJ745450	KY214185	–	KY241290	KY241254	KY241269	–
*A. mccleeryorum* (WAM T53979)	MT611168	MT603527	MT604146	MT623663	MT604125	MT604135	–
*A. mellosa* (WAM T107182)	KJ745440	KY214184	KY241237	MG800294	KY241253	KY241268	MG800231
*A. pallida* (QMB S86817)	KY241278	KY214179	KY241230	KY241283	–	–	–
*A. phillipae* (WAM T110142)	–	–	MT604160	MT623668	MT604128	MT604141	MT623659
*A. salina* (WAM T148204)	OQ922000	OQ925882	OQ925872	OQ924319	OQ925885	OQ925881	OQ924318
*A. simoneae* (WAM T110261)	MT611175	–	MT604158	MT623669	MT604129	MT604144	MT623660
*A. sinuata* (WAM T129020)	MN635073	MN634957	MN634758	MN635099	MN634924	MN634745	MN635131
*A. vernonorum* (WAM T98767)	MG800161	MG799887	MG799953	MG800290	MG800025	MG800102	MG800226
*A. watsoni* (WAM T96018)	KJ745407	MN634960	MN634776	MN635125	MN634929	MN634750	–
*A. whitei* (WAM T127202)	KJ745174	–	MN634900	MN635108	MN634928	MN634726	MN635134
*A.* sp. ‘Goodnight’ (QMB S111402)	–	–	MT281027	MT281268	MT281106	MT281182	MT281249
*A.* sp. ‘Goodnight’ (QMB S111405)	MT280892	–	MT281026	MT281267	MT281105	MT281181	MT281248
*A.* sp. ‘Paluma’ (QMB S111473)	–	MT280963	MT281028	MT281266	MT281104	MT281180	–
***Chenistonia* Hogg, 1901 [3]**							
*C.* sp. ‘MYG348′ (WAM T72687)	KJ745221	KY214180	KY241231	KY241284	KY241247	KY241262	MG800196
*C.* sp. ‘MYG348′ (WAM T81017)	MG800151	MG799878	MG799940	MG800276	MG800011	MG800088	MG800210
*C.* sp. ‘MYG348′ (WAM T81018)	MG800152	MG799879	MG799941	MG800277	MG800012	MG800089	MG800211
***Hesperonatalius* Castalanelli, Huey, Hillyer & Harvey, 2017 [2]**							
*H. langlandsi* (WAM T108988)	KJ744689	KY214189	KY241243	–	KY241258	KY241274	MG800232
*H. maxwelli* (WAM T108989)	KJ744690	KY214190	KY241244	KY241293	KY241259	KY241275	MG800233
***Kwonkan* Main, 1983 [11]**							
*K. turrigera* (WAM T134203)	MG800182	MG799911	–	MG800313	MG800056	MG800134	MG800254
*K. wonganensis* (WAM T157120)	OQ922001	OQ925883	OQ925873	OQ924322	OQ925884	OQ925877	OQ924317
*K.* sp. ‘MYG088’ (WAM T54237)	KJ745206	MG799864	MG799923	MG800267	MG799994	MG800071	MG800191
*K.* sp. ‘MYG165’ (WAM T99672)	KJ745512	KY214187	KY241239	KY241292	KY241256	KY241271	MG800229
*K.* sp. ‘MYG195’ (WAM T132749)	MG800177	MG799906	–	MG800308	MG800049	MG800127	MG800250
*K*. sp. ‘MYG197’ (WAM T130375)	MG800166	–	MG799963	MG800301	MG800038	MG800116	MG800239
*K*. sp. ‘MYG343’ (WAM T57563)	KJ745210	MG799866	MG799925	–	MG799996	MG800073	MG800193
*K.* sp. ‘MYG390’ (WAM T132361)	MG800173	MG799902	MG799970	MG800304	MG800045	MG800123	MG800247
*K*. sp. ‘MYG392’ (WAM T132363)	MG800175	MG799904	MG799972	MG800306	MG800047	MG800125	–
*K*. sp. ‘MYG458’ (WAM T88514)	MG800155	MG799880	MG799944	MG800280	MG800015	MG800092	MG800213
*K.* sp. ‘MYG650’ (WAM T145316)	MT656266	MT656258	–	–	–	–	–
***Namea* Raven, 1984 [9]**							
*N. brisbanensis* (QMB S111356)	MT280866	MT280912	MT281000	–	MT281078	MT281155	MT281223
*N. flavomaculata* (WAM T133310)	KY241282	KY214192	KY241246	KY241294	KY241261	KY241277	MG800253
*N. salanitri* (QMB S111396)	MT280830	MT280924	MT280977	–	MT281057	MT281119	MT281224
*N. dahmsi* (QMB S111381)	MT280891	–	MT281022	MT281262	MT281056	MT281173	MT281244
*N. jimna* (QMB S111410)	MT280875	–	MT281011	–	MT281093	MT281132	MT281236
*N. excavans* (QMB S111535)	MT280884	MT280941	MT281025	–	–	–	–
*N.* sp. ‘Goomboor.’ (QMB S11135)	MT280881	MT280950	MT281013	MT281261	MT281089	MT281139	MT281238
*N.* sp. ‘Kroombit’ (QMB S111330)	MT280887	MT280960	–	–	MT281051	MT281171	MT281231
*N.* sp. ‘Ravensb.2’ (QMB S111445)	MT280883	MT280942	MT281019	MT281264	MT281043	MT281175	MT281228
***Proshermacha* Simon, 1908 [9]**							
*P*. sp. ‘MYG344’ (WAM T132981)	MG800181	–	MG799978	MG800311	MG800054	MG800132	MG800251
*P*. sp. ‘MYG346’ (WAM T80952)	MG800150	–	MG799939	MG800275	MG800010	MG800087	MG800209
*P*. sp. ‘MYG349’ (WAM T72701)	KJ745222	–	MG799928	–	MG799999	MG800076	MG800197
*P*. sp. ‘MYG357’ (WAM T78535)	KY241279	–	KY241232	KY241285	KY241248	KY241263	MG800206
*P*. sp. ‘MYG465’ (WAM T132960)	MG800180	–	MG799977	–	MG800053	MG800131	–
*P*. sp. ‘MYG467’ (WAM T131982)	MG800169	–	MG799966	–	MG800041	MG800119	MG800242
*P*. sp. ‘MYG468’ (WAM T96060)	MG800158	–	MG799948	MG800285	MG800020	MG800097	MG800220
*P*. sp. ‘MYG469’ (WAM T94765)	MG800156	–	MG799945	MG800281	MG800016	MG800093	MG800214
*P*. sp. ‘MYG471’ (WAM T132903)	–	MG799907	MG799974	MG800309	MG800050	MG800128	–
***Swolnpes* Main & Framenau, 2009 [4]**							
*S. darwini* (WAM T97003)	KY241280	KY214183	KY241236	KY241289	KY241252	KY241267	MG800223
*S. darwini* (WAM T97503)	MG800160	MG799884	MG799950	MG800287	MG800022	MG800099	MG800224
*S.* sp. ‘MYG234’ (WAM T114056)	–	MG799894	–	–	MG800032	MG800109	–
*S*. sp. ‘MYG415’ (WAM T53579)	MG800145	MG799863	–	MG800266	MG799993	MG800070	–
***Teyl* Main, 1975 [9]**							
*T. damsonoides* (WAM T137482)	MG800187	MG799915	MG799984	–	MG800061	–	MG800259
*T. heuretes* (WAM T91918)	MN101151	MN104643	MN104652	–	MN104658	MN104650	MN101153
*T. luculentus* (WAM T141133)	MN101147	–	–	–	–	–	–
*T. meridionalis* (WAM T147625)	MT280814	MT280897	MT280969	–	MT281039	MT281115	MT281188
*T. vancouveri* (WAM T16804)	–	MN104644	MN104653	–	–	–	–
*T*. sp. ‘MYG012’ (WAM T96062)	KJ745412	KY214182	KY241235	KY241288	KY241251	KY241266	MG800221
*T*. sp. ‘MYG053’ (WAM T96326)	MG800159	MG799883	MG799949	MG800286	MG800021	MG800098	MG800222
*T*. sp. ‘MYG358’ (WAM T78529)	KJ745277	MG799875	MG799936	MG800274	MG800007	MG800084	MG800205
*T*. sp. ‘MYG412’ (WAM T116018)	MG800164	MG799895	MG799961	MG800297	MG800034	MG800111	MG800189
***Teyloides* Main, 1985 [1]**							
*T. bakeri* (SAM NN29525)	MG800144	MG799861	–	MG800265	MG799991	MG800068	MG800190
***Troglodiplura* Main, 1969 [4]**							
*T. beirutpakbarai* (AM KS30823)	–	–	MT656259	–	–	–	–
*T. beirutpakbarai* (SAM NN30808)	OQ922002	–	OQ925874	–	OQ925886	OQ925878	OQ924314
*T. beirutpakbarai* (SAM NN30809)	OQ922003	–	OQ925875	OQ924320	OQ925887	OQ925880	OQ924315
*T. beirutpakbarai* (SAM NN30810)	OQ922004	–	OQ925876	OQ924321	OQ925888	OQ925879	OQ924316
**DIPLURIDAE** ***Linothele* Karsch, 1879 [1]**							
*L.* sp. indet. (#ZSMA20170069)	–	MG273475	MG273475	MG273544	MG273579	MG273624	–
**MICROSTIGMATIDAE** ***Ixamatus* Simon, 1887 [1]**							
*I.* sp. indet. (#ARAMY002102)	KY017807	–	KY015981	KY018321	DQ639831	KY017212	JQ358747
***Kiama* Main & Mascord, 1969 [1]**							
*K. lachrymoides* (#ARAMY002094)	–	–	KY015982	KY018322	DQ639796	DQ639884	JQ358748
***Xamiatus* Raven, 1981 [2]**							
*X. rubrifrons* (QMB S111351)	MT656265	–	MT656260	MT656268	MT656262	MT656264	–
*X.* sp. indet. (QMB S111284)	–	–	–	MT656267	MT656261	MT656263	–
**PYCNOTHELIDAE** ***Acanthogonatus* Karsch, 1880 [1]**							
*A. campanae* (#ARAMYCAS539)	KY017803	–	KY015977	–	DQ639843	KY017208	JQ358720
***Stanwellia* Rainbow & Pulleine, 1918 [4]**							
*S. nebulosa* (SAM NN28449)	MG800186	MG799858	MG799983	–	MG800060	MG800138	MG800258
*S.* sp. indet. (QMB S111338)	MT280818	–	MT281031	–	MT281096	MT281178	MT281251
*S.* sp. indet. (QMB S111388)	–	–	MT281030	–	MT281095	MT281177	–
*S.* sp. indet. (QMB S111428)	MT280817	–	MT281029	MT281252	MT281094	MT281176	MT281250

## Data Availability

The data presented in this study are available in Appendix A.

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
