# Peer review of "A Case for Below-Ground Dispersal? Insights into the Biology, Ecology and Conservation of Blind Cave Spiders in the Genus Troglodiplura (Mygalomorphae: Anamidae)"

_insects, 2023, doi:10.3390/insects14050449_

Round 1
Reviewer 1 Report
Dear authors
Thank you for this interesting and well prepared study. I have only some minor comments.
Best

Author Response
We thank Reviewer 1 for their thorough and constructive comments on this manuscript, please find below our responses to the reviewer's comments:
Reviewer: 1
Reviewer Comments to the Author:
Point 1: Line 75– 77 I suggest to put this in the reference section and cite here in the usual matter.
Response 1: We have made this suggested edit.
Point 2: Line 170 – 173 Unfortunately, the map labelling is not legible, and there is also no overview map to better classify the study area.
Response 2: We have made the suggested edit – changing the basemap to one without text, italicising the font in the legend, and adding an overview map.
Point 3: Line 179 Is this your nomenclature? Which one is it on the map?
Response 3: This is an Australian Speleological Federation number, a sentence has been added to the end of section 2.1 to clarify this. Reference has been added to Figure 1 to indicate the location of N-253.
Point 4: Line 187 – 189 I assume the photo shows the entrance? Please state this also in the description. Could you please also indicate the size of the entrance? It is really difficult to see the extension.
Response 4: We have added a sentence to the caption to explain that this is an image of the cave entrance, with the approximate proportions of the entrance.
Point 5: Line 190 Did you visit every cave once or several times? How long did you survey the respective caves?
Response 5: We have added a paragraph to the methods section 2.3 Survey Methods to clarify this.
Point 6: Line 199 – 200 Can you please be more specific. Does that mean you did not collect more than 1?
Response 6: We have added a paragraph to section 2.3 Survey methods to describe the number of specimens collected.
Point 7: Line 357 – 358 In my opinion this map/figure can be deleted, because it doesn't add any valuable information
Response 7: This figure shows the relative position of the caves in which Troglodiplura beirutpakbarai were found and so we believe it adds information by clarifying the relation of the caves to each other. We have edited the image and caption to better show and explain this.
Point 8: Line 363 A scale would be great. Can you give a rough estimate of body size and leg length.
Response 8: We have replaced Figure 5c to include a scale bar. We have added a section to describe the morphology 3.3 Intraspecific Variation.
Point 9: Line 380 – 382 This rather part of the discussion section
Response 9: Thank you, this has been moved to the discussion.
Point 10: Line 384 Caption and figure should fit on one page and blue is not the best of choices to highlight something
Response 10: We have edited the image so that the image and caption fit on one page. We disagree about the comment about the colour choices, and choose to retain the chosen colour scheme.
Point 11: Line 404 Please refer to the yellow arrows. I can't identify mud holes in photo a). I assume the scale on the left indicates that the photo is ~7cm width?
Response 11: The yellow arrows have been edited to improve clarity and highlight the mud holes. A sentence has been added to describe the scale.
Point 12: Line 452 – 456 Discussion
Response 12: We have moved down this paragraph and incorporated it in the discussion
Point 13: Line 461- 462 Or emphasizes the limitations of COI for some groups of spiders.
Response 13: We disagree. The limitations of COI come into play when high mitochondrial divergence values indicate speciation (using single locus delimitation methods), when in fact nuclear data indicate otherwise. COI is still an extremely useful and informative gene for indicating conspecificity and gene flow when divergence values are very low (as here).
Point 14: Line 476 – 506 To me this (incl. Figure 7) would make more sense in the methods or the results section
Line 507 – 531 This should be in the methods section in the description of the study area. In my opinion your study doesn't add so much about the morphology of the caves that it should be part of the discussion section. If you want this part in the discussion section you should clearly state how this relates to your findings on the troglobiont species.
Response 14: We have moved this section, plus figure 7 up to the methods, under Section 2.1, Study Area. We have also merged the arrows into an individual image.
Point 15: Line 564: Reference?
Response 15: We have added references as suggested.
Point 16: Line 574: Reference?
Response 16: We have added references as suggested.
Point 17: Lines 600-603: Methods
Response 17: We have incorporated this paragraph into the methods, section 2.2, Site Selection
Point 18: Line 616: Have you considered eDNA?
Response 18: We have added a discussion of eDNA to this paragraph
Reviewer 2 Report
This is a very nice manuscript, reporting the interesting result that several (seemingly) disjunct cave populations of an endemic diplurid are conspecific and provide evidence that these caves were all previously interlinked. The molecular data is consistent with what is known for the phylogeny of this and related groups. I have one minor corrections: Line 64: no need for e.g. in square bracket reference. I would also consider it informative if the authors can at least briefly comment on the morphological variation of these populations. As much mygalomorph systematics is still reliant on morphology it would be useful to state whether they see strong homogeneity or if there is any noticeable intraspecific variation. This can be achieved in one extra paragraph in the text.
Author Response
Referee: 2
Comments to the Author:
Point 1: This is a very nice manuscript, reporting the interesting result that several (seemingly) disjunct cave populations of an endemic diplurid are conspecific and provide evidence that these caves were all previously interlinked. The molecular data is consistent with what is known for the phylogeny of this and related groups.
Response 1: We thank you for these positive comments.
Point 2: I have one minor corrections: Line 64: no need for e.g. in square bracket reference.
Response 2: This has been amended.
Point 1: I would also consider it informative if the authors can at least briefly comment on the morphological variation of these populations. As much mygalomorph systematics is still reliant on morphology it would be useful to state whether they see strong homogeneity or if there is any noticeable intraspecific variation. This can be achieved in one extra paragraph in the text.
Response 2: Thank you for this comment, we have added a paragraph in the results to describe morphological variation; 3.3 Intraspecific variation.
Reviewer 3 Report
This is an excellent work, precisely covering the diverse issues with cave inhabiting species and, indeed, challenging the species concept in these remarkable animals: why no genetic divergence over such distances and time? Perhaps the organisms subjected to a highly conserved environment are under no selection pressure. I found it read well but in at least two instances attention can be given to English and punctuation:
line 583 begins a paragraph covering diverse issues. Consideration should be given to making more paragraphs.
line 611 "This study provides important baseline data for these spiders, ?STOP/colon/semicolon? however a large number of caves of the Nullarbor Plain are yet to have had biological surveys and the potential
for them to harbor similarly rare, endemic, and vulnerable biota is high."
On the taxon level, Atypus snetsingeri is now considered a synonym of A. karschi, on molecular grounds. On that issue, I did wonder why the non-anamid taxa are listed but not figured with the molecular summary.
Another superb piece of work from this team. Congratulations.
Excellent with the above concerns
Author Response
Reviewer 3.
Comments to the author:
Point 1: This is an excellent work, precisely covering the diverse issues with cave inhabiting species and, indeed, challenging the species concept in these remarkable animals: why no genetic divergence over such distances and time? Perhaps the organisms subjected to a highly conserved environment are under no selection pressure.
Response 1: Thank you for these positive comments, the results do open up many more intriguing questions.
Point 2: I found it read well but in at least two instances attention can be given to English and punctuation:
line 583 begins a paragraph covering diverse issues. Consideration should be given to making more paragraphs.
Response 2: Thank you for this comment. This paragraph discusses the range of threats potentially acting on the species and / cave environ, hence it was grouped together, we have no edited to separate out some sections
Point 3: line 611 "This study provides important baseline data for these spiders, ?STOP/colon/semicolon? however a large number of caves of the Nullarbor Plain are yet to have had biological surveys and the potential for them to harbor similarly rare, endemic, and vulnerable biota is high."
Response 3: We have added a STOP where suggested.
Point 4: On the taxon level, Atypus snetsingeri is now considered a synonym of A. karschi, on molecular grounds. On that issue, I did wonder why the non-anamid taxa are listed but not figured with the molecular summary.
Response 4: We have edited table accordingly. Re. the outgroups, they are excluded from the figure as this paper is only concerned with Anamidae; they were included in the molecular dataset as it was an expanded re-analysis of a previous study. Thus, they are of no relevance to the presented phylogeny or results as they pertain to Troglodiplura.